# The Potential of *Trigona* spp. Propolis as an Antioxidant Agent to Reduce Residual Peroxide after Intra-Coronal Bleaching Treatments

**Aries Chandra Trilaksana** [1,*]**, Syamsiah Syam** [2] **, Muhammad Ruslin** [3] **and Yung-Kang Shen** [4]

1    Department of Conservative Dentistry, Faculty of Dentistry, Hasanuddin University, Makassar 90245, Indonesia

2    Faculty of Dentistry, Universitas Muslim Indonesia, Makassar 90231, Indonesia; syams_77@umi.ac.id

3    Department of Oral Maxillofacial Surgery, Faculty of Dentistry, Hasanuddin University, Makassar 90245, Indonesia; mruslin@unhas.ac.id

4    School of Dental Technology, College of Oral Medicine, Taipei Medical University, Taipei 110, Taiwan; ykshen@tmu.edu.tw

*    Correspondence: arieschandra@unhas.ac.id

**Abstract:** The present study aimed to determine the effectiveness of *Trigona* spp. propolis as an antioxidant to reduce residual peroxide after intra-coronal bleaching treatments. Thirty-five maxillary central incisors were divided into seven groups: five samples without antioxidants; sodium ascorbate 10% combined with Tween 80 0.2%; and *Trigona* spp. propolis 10%. The lengths of the application time were 1 h, 24 h, and 48 h. Each application time consisted of five samples. Root resection followed by artificial discoloration was performed in the samples. Then, intra-coronal bleaching using 35% hydrogen peroxide was applied. After the tooth color changed, the bleaching material was cleared, and this was followed by the applications of sodium ascorbate 10% combined with Tween 80 0.2% and *Trigona* spp. propolis 10%. The peroxide residue was measured by assessing dissolved oxygen using a titration analysis with either the Winkler or iodometric method. Data were analyzed using the ANOVA test and Tukey's HSD test. The lowest peroxide residue amount was found with the application of antioxidants for 48 h after the intra-coronal bleaching treatment using 35% hydrogen peroxide. However, there was no significant difference between sodium ascorbate 10% combined with Tween 80 0.2% and Trigona spp. propolis 10% to reduce peroxide residues after the intra-coronal bleaching treatment ($p > 0.05$). Therefore, these findings indicate that *Trigona* spp. propolis 10% effectively reduces peroxide residues after intra-coronal bleaching treatments, which can interfere with the bond of the composite to the tooth surface and shorten the wait time for composite restorations after bleaching treatments.

**Keywords:** *Trigona* spp. propolis; residual peroxide; intra-coronal bleaching





## 1. Introduction

Tooth discoloration is a change in hue, color, or dental translucency due to extrinsic and intrinsic factors [1–4]. Intrinsic discoloration can be caused by pulp necrosis, intra-pulpal hemorrhage, dentin hyper-calcification, age, defects during tooth formation, drug influence during root canal treatment, and restorative materials (amalgam, pin, wedge, and composite) [4–9]. Crowns, labial veneers, and intra-coronal bleaching are all options for treating intrinsically discolored teeth [9,10]. These treatments can change the color of discolored teeth such that they become bright. Due to globalization and the growing modernization of all sectors, both men and women have developed heightened aesthetic sensibilities and place a premium on any aspect of their look that would improve their appearance, including smiling brightly and whitely [11–14]. Because tooth whitening is an affordable, quick, and minimally invasive process, it is now the treatment of choice for tooth staining among all available alternatives for treating tooth discoloration [11,12].

Intra-coronal bleaching is a method of whitening nonvital teeth after root canal treatment and involves placing strong oxidizing agents in the pulp chamber [10,15]. Bleaching agents include hydrogen peroxide, sodium perborate, sodium hypochlorite, and carbamide peroxide. However, in the case of intra-coronal bleaching, 10–15% sodium perborate and 35% hydrogen peroxide are often used [1,4,6,7,15].

Hydrogen peroxide has a clear liquid form and is highly unstable, odorless, and acidic [6–8,16]. This bleaching material is a strong oxidizer and is easily mixed with water, alcohol, or ether [4,7]. The oxidation reactions of peroxides produce gaseous bubbles under the enamel surface. The oxygen-containing surface layer that inhibits the polymerization of bonding agents affects the strength of enamel–resin bonds [6,10,17,18]. Various studies suggest suspending the restoration procedure for 2–4 weeks to prevent bond failure between enamel and composite restorations [18]. However, this poses a clinical problem when the tooth should be immediately restored with a composite restoration. Therefore, a material that can accelerate the loss of peroxide residue is required [6,10,17,19].

Antioxidants, such as vitamin C (ascorbic acid), glutathione, beta carotene, vitamin E, CoQ10, flavonoids, and lipoic acid, are able to neutralize hydrogen peroxide residue and the oxygen-rich molecule [20–24].

Propolis is a natural antioxidant source and is relatively easier to obtain than synthetic antioxidants such as sodium ascorbate [24,25]. Propolis, or bee glue, is a resin material collected by honeybees from various plant species [25–27]. A species of bee that can produce propolis in large quantities is *Trigona* spp. This type of bee is commonly found in South Sulawesi, both in the highlands and lowlands. Propolis has vigorous antioxidant activity against oxidants and free radicals ($H_2O_2$, $O_2$-, and OH) compared with other bee products [28,29]. The matrix of flavonoids in propolis can reduce the harmful effects of free radicals. Therefore, the present study aimed to assess the potential of propolis *Trigona* spp. as an antioxidant agent to reduce residual peroxide after an intra-coronal bleaching treatment. Accordingly, patients could obtain esthetic restoration immediately after the intra-coronal bleaching treatment.

## 2. Materials and Methods

### 2.1. Material Extraction

*Trigona* spp. propolis from Bulukumba district of South Sulawesi was extracted with the maceration method using 95% ethanol diluent with the following process: 840 g of *Trigona* spp. propolis was put into a maceration container; then, we added 1000 mL of 95% ethanol and stored it in a place protected from direct sunlight and allowed it to stand for 24 h while stirring occasionally. Furthermore, the propolis extract was filtered, the filtrate was collected, and the dregs were re-extracted with 95% ethanol diluent in the same way as before for 3 × 24 h. The obtained filtrate was combined, and then the solvent was evaporated using a rotary evaporator to produce a thick extract. The extract was stored in a desiccator until the test was carried out.

### 2.2. Tooth Staining Process

The current study used 35 extracted maxillary incisors that were extracted due to periodontal disease. Debris and calculus were removed using an ultrasonic scaler. Teeth was soaked in distilled water. The cavity access was made by using carbide bur no. 2 with a high-speed handpiece, and a third cervical form of the root canal was dilated with Gates Glidden drill no. 2. Roots that were resected between the coronal and middle third (1 mm to apical against the cementoenamel junction) were measured by the probe, using a carborundum disk. The crown was immersed in 17% Ethylene Diamine Tetra-acetic Acid (EDTA) for 5 min to remove the smear layer. Then, teeth were washed with distilled water.

Artificial discoloration was performed using the Freccia and Peters method whereby the tooth was immersed in cow blood without serum and centrifuged at 3200 rpm for 20 min (twice a day for about three days) to increase penetration of hemolyzed blood cells into the dentinal tubules. The precipitate was removed, and the tooth was immersed in

the remaining hemoglobin-rich hemolysate for three days. Centrifugation was performed twice a day, as described previously. The tooth, which had changed in color, was washed in distilled water.

*2.3. Intra-Coronal Bleaching Treatment and Dissolved Oxygen Analysis*

The cervical discolored tooth was covered with a glass ionomer cement (Glass Ionomer Cement Fuji IX Extra, GC Corporation, Tokyo, Japan). The thickness of glass ionomer cement was 2 mm; it was placed 1 mm below the cementoenamel junction to obtain apical closure. A shade guide was used to evaluate the initial colors of discolored teeth, and the tooth was treated with intra-coronal bleaching using 35% hydrogen peroxide (Opalescence®Endo, Ultradent Products, Inc., South Jordan, UT, USA). Thereafter, the tooth color was determined.

The samples were divided into several groups based on the type and duration of antioxidant application: without antioxidants group as control group (5 samples), sodium ascorbate 10% combined with Tween 80 0.2% group (5 samples, 1 h), sodium ascorbate 10% combined with Tween 80 0.2% group (5 samples, 24 h), sodium ascorbate 10% combined with Tween 80 0.2% group (5 samples, 48 h), *Trigona* spp. propolis 10% group (5 samples, 1 h), *Trigona* spp. propolis 10% group (5 samples, 24 h), and *Trigona* spp. propolis 10% group (5 samples, 48 h).

After the intra-coronal bleaching procedure was completed, the pulp chamber was irrigated with 2 mL of aquades. Antioxidant materials from sodium ascorbate 10% combined with Tween 80 0.2% and *Trigona* spp. propolis 10% were placed into the pulp chamber and then covered with the temporary restoration, which was stored in a glass tube containing NaCl. After 1 h, 24 h, and 48 h, samples of each group were issued and rinsed with water. Samples were inserted in a Hale tube, and the dissolved oxygen titration was ready to be analyzed using either Winkler or iodometric method with triplicate reading.

*2.4. Statistical Analysis*

All the results of laboratory observations were collected and processed with SPSS software version 17.0 (SPSS Inc., Chicago, IL, USA), and they were analyzed using the one-way ANOVA test and Tukey's HSD test. A value of $p < 0.05$ indicated statistical significance.

## 3. Results

This research was experimental and was conducted in a laboratory. The dependent variable was peroxide residue at 1 h, 24 h, and 48 h, while *Trigona* spp. propolis 10% and Sodium Ascorbate 10% + Tween 80 0.2% were the independent variables. The Shapiro–Wilk normality test gave a $p$-value $> 0.05$, suggesting that the data were normally distributed. Figure 1 presents the average dissolved oxygen (ppm) results. The statistical test, a one-way ANOVA, found a $p$-value $< 0.05$ in both antioxidant groups, i.e., *Trigona* spp. propolis 10% and sodium ascorbate combined with Tween 80 0.2%. These results indicated a significant difference in peroxide residue between *Trigona* spp. propolis 10% and sodium ascorbate combined with Tween 80 0.2% in the application period.

Table 1 shows the results of different advanced tests that we used to analyze the results that showed the effects of different application duration times, i.e., 1 h, 24 h, and 48 h, in each antioxidant group. In the *Trigona* spp. propolis 10% antioxidant group, we found a significant difference in peroxide residue between 1 h and 24 h that was 1320 ppm ($p = 0.000$; $p < 0.05$). The results also showed that there was a significant difference ($p = 0.000$, $p < 0.05$) of 1020 ppm between 24 h and 48 h. There was a difference between 1 h and 48 h ($p = 0.000$, $p < 0.05$) as well. In sodium ascorbate 10% combined with Tween 80 0.2%, a significant difference ($p = 0.000$, $p < 0.05$) was found between 1 h and 24 h, which was 1940 ppm. Between 24 h and 48 h, there was a significant residual difference ($p = 0.000$, $p < 0.05$) of 0.876 ppm. There was also a significant difference between 1 h and 48 h ($p = 0.000$, $p < 0.05$).

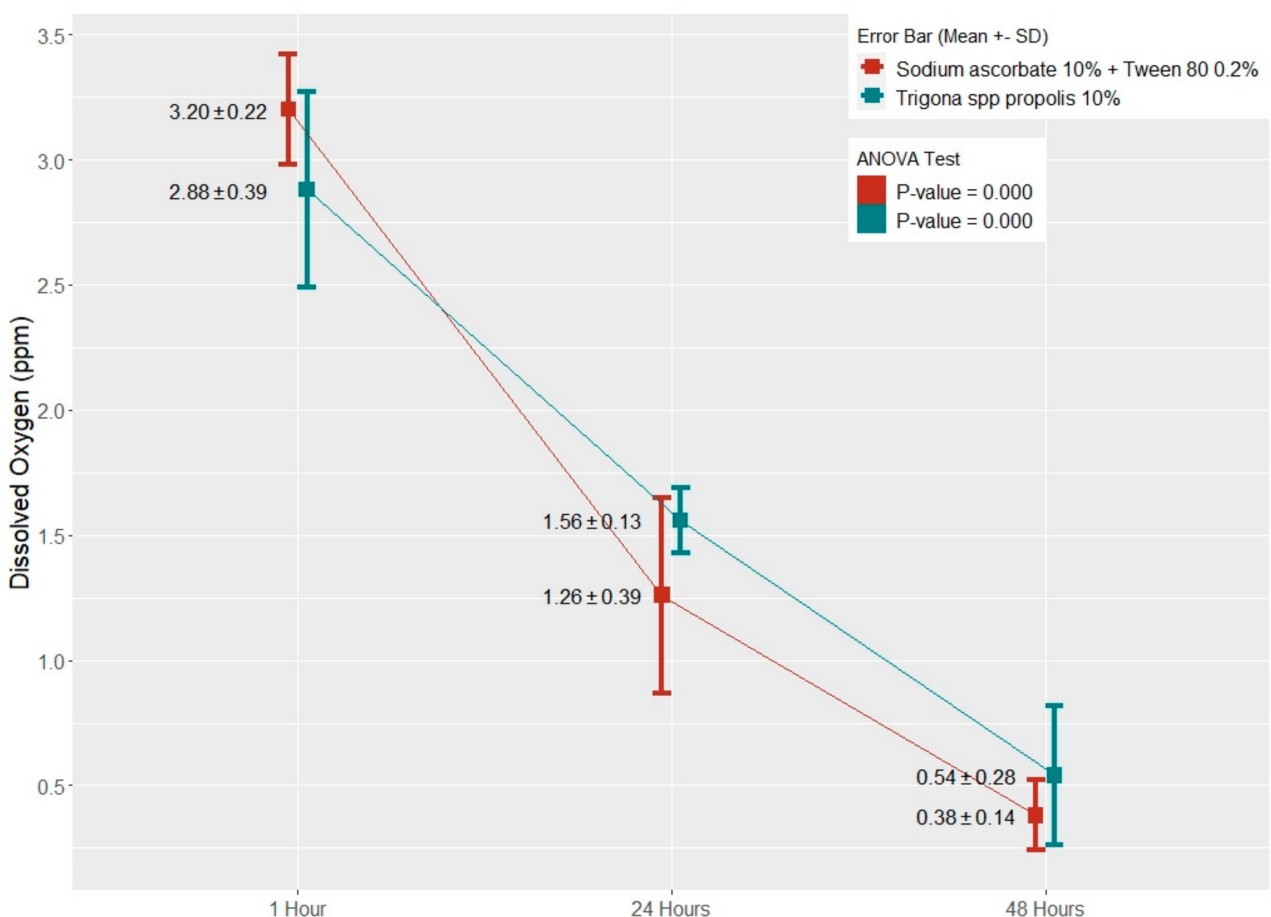

**Figure 1.** The average distribution of dissolved oxygen based on duration of application time in each antioxidant group. (The difference between the average dissolved oxygen of each antioxidant group during application time was statistically significant at 0.05 level.)

**Table 1.** The test results for application duration times of 1 h, 24 h, and 48 h in each antioxidant group.

| Antioxidant | The Length of Application Time | Comparison | Mean Difference | 95% CI (Min–Max) | *p*-Value |
|---|---|---|---|---|---|
| *Trigona* spp. propolis 10% | 1 h | 24 h | 1.320 | 0.832–1.807 | 0.000 * |
| | | 48 h | 2.340 | 1.852–2.827 | 0.000 * |
| | 24 h | 48 h | 1.020 | 0.532–1.507 | 0.000 * |
| Sodium Ascorbate 10% + Tween 80 0.2% | 1 h | 24 h | 1.940 | 1.478–2.401 | 0.000 * |
| | | 48 h | 2.816 | 2.354–3.277 | 0.000 * |
| | 24 h | 48 h | 0.876 | 0.414–1.337 | 0.001 * |

* Post hoc test: Tukey's HSD test; $p < 0.05$: significant.

Figure 2 shows significant differences in peroxide residues among the *Trigona* spp. propolis 10% group, sodium ascorbate 10% combined with Tween 80 0.2% group, and control group ($p = 0.000$; $p < 0.05$) for each application duration time.

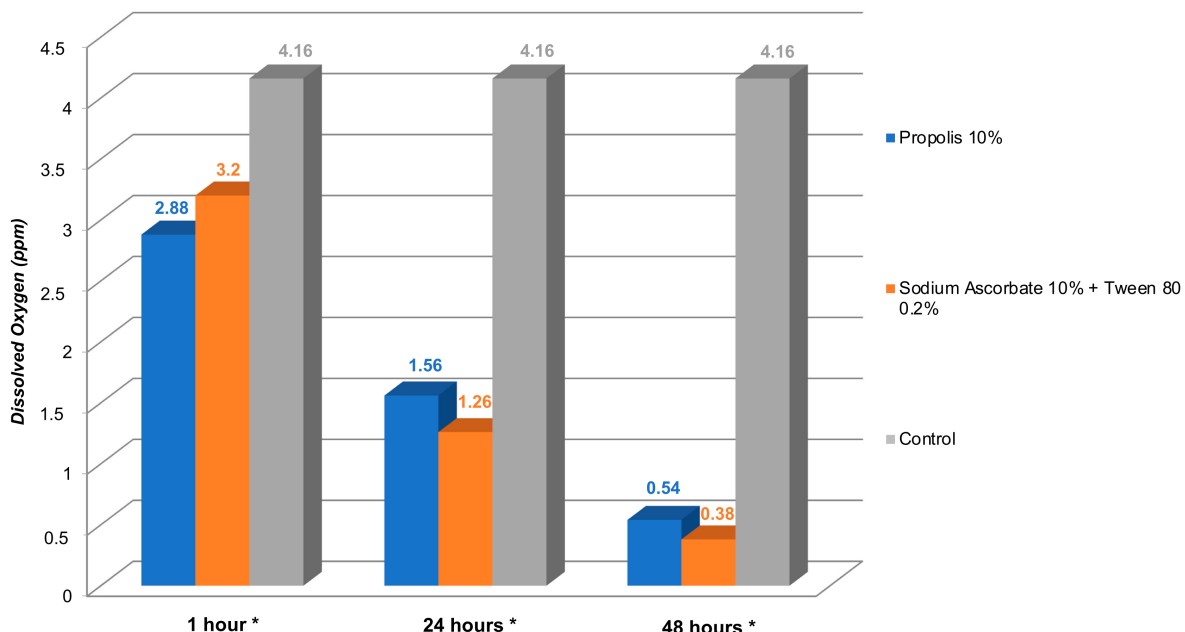

**Figure 2.** The average distribution of dissolved oxygen in the antioxidant and control groups with different lengths of application, i.e., 1 h, 24 h, and 48 h. (* Difference between average of dissolved oxygen in each antioxidant group for 1 h, 24 h, and 48 h was statistically significant at 0.05 level).

To see further group differences, a post hoc test was conducted. Table 2 shows the results of the advanced difference test between the *Trigona* spp. propolis 10% group, sodium ascorbate 10% combined with Tween 80 0.2% group, and control group with different application times of 1 h, 24 h, and 48 h. There was a significant difference between *Trigona* spp. propolis 10% and the control group. The same result was also found between the sodium ascorbate 10% combined with Tween 80 0.2% group and the control group. However, no significant difference was found between *Trigona* spp. propolis 10% and sodium ascorbate 10% combined with Tween 80 0.2%.

**Table 2.** The difference test results showing comparisons among antioxidant groups, i.e., *Trigona* spp. propolis 10%, sodium ascorbate 10% combined with Tween 80 0.2%, and control group, with different lengths of application, i.e., 1 h, 24 h, and 48 h.

| The Length of Application Time | Antioxidant | Comparison | Mean Difference | 95% CI (Min–Max) | *p*-Value |
|---|---|---|---|---|---|
| 1 h | *Trigona* spp. propolis 10% | Sodium Ascorbate 10% + Tween 80 0.2% | −0.320 | −0.98–0.34 | 0.426 |
| | | Control | −1.280 | −1.94−−0.61 | 0.001 * |
| | Sodium Ascorbate 10% + Tween 80 0.2% | Control | −0.960 | −1.62−−0.29 | 0.006 * |
| 24 h | *Trigona* spp. propolis 10% | Sodium Ascorbate 10% + Tween 80 0.2% | 0.300 | −0.33–0.93 | 0.444 |
| | | Control | −2.600 | −3.23−−1.96 | 0.000 * |
| | Sodium Ascorbate 10% + Tween 80 0.2% | Control | −2.900 | −3.53−−2.26 | 0.000 * |
| 48 h | *Trigona* spp. propolis 10% | Sodium Ascorbate 10% + Tween 80 0.2% | 0.156 | −0.42–0.73 | 0.758 |
| | | Control | −3.620 | −4.20−−3.03 | 0.000 * |
| | Sodium Ascorbate 10% + Tween 80 0.2% | Control | −3.776 | −4.35−−3.19 | 0.000 * |

* Post hoc test: Tukey's HSD test; *p* < 0.05: significant.

## 4. Discussion

Tooth discoloration from the intra-coronal bleaching process occurs due to the diffusion of free radicals, such as hydroxyl, superoxide anion, oxygen, and per hydroxyl, produced through the oxidation–reduction reaction of hydrogen peroxide [6,10,30–33]. This oxygen-rich molecule creates peroxide residues in gaseous bubbles that form under the enamel, which can inhibit and prevent adequate infiltration of the bonding agent into the tooth structure [6,10,17,33,34].

This study showed that all the study groups showed the presence of peroxide residues characterized by the release of oxygen or dissolved oxygen. The high dissolved oxygen in the control group (without antioxidants) was due to the large amount of residual peroxide in the dentin and enamel. In line with our study, de Oliveira et al. (2011) stated that bleaching teeth releases significant amounts of oxygen 24 h after treatment [30]. Studies have also recommended postponing the bonding procedures when bleaching teeth until the remaining oxygen on the teeth surfaces can be removed [31,33,35–37].

Dissolved oxygen in the sodium ascorbate 10% combined with Tween 80 0.2% group and the *Trigona* spp. propolis 10% group was lower than that in the control group (without antioxidants) because both materials have antioxidant activity. Antioxidants are substances that neutralize free radicals by emitting their electrons, which stop the reaction that causes electron loss [35,38–41]. The combination of Tween 80 0.2% with sodium ascorbate causes surface tension and affects the contact angle. In addition, the cohesion force between sodium ascorbate molecules is reduced so that it is easier for sodium ascorbate to penetrate the dentin [42].

*Trigona* spp. propolis produced via selected extraction has a brown-to-dark-brown color with a soft consistency. Dark-colored propolis contains abundant flavonoids, so it generates more flavonoids than other types [43,44]. This flavonoid is an active substance in propolis. Moreover, color is a marker of active substance levels in an object or chemical substance in materials [43,45].

In this study, dissolved oxygen in the *Trigona* spp. propolis group was lower than that of the control group (without antioxidants) because *Trigona* spp. propolis extracted using ethanol showed moderate antioxidant activity. Ethanol propolis extract (EEP) has remarkable medical benefits, namely, protection against gamma radiation [46,47]. The anti-oxidative effects of EEP are associated with free radical absorbability [46–48]. A study also showed that increasing the amount of EEP inhibits the luminol of $H_2O_2$ chemiluminescence in an in vitro way and suggested that its anti-oxidative capacity is due to the high flavonoid substance [49].

The results of the post hoc test, Tukey's HSD test ($p < 0.05$: significance), comparing the observation times of 1 h, 24 h, and 48 h for each antioxidant group suggested that the duration of the antioxidant application affected the peroxide residue, as indicated by a significant decrease in dissolved oxygen at 1 h, 24 h, and 48 h. Sodium ascorbate 10% combined with Tween 80 0.2% showed lower dissolved oxygen than *Trigona* spp. propolis 10% with 48 h of application. Nevertheless, there was no statistically significant difference for both groups. This result is in line with the research conducted by Yusri et al. (2016), who showed that sodium ascorbate 10% combined with Tween 80 0.2% has the lowest coronal microleakage after intra-coronal bleaching using 35% hydrogen peroxide and requires a minimum application time of 48 h before composite restorations [50]. Regardless of the peroxide residue, several studies have shown that short application times (2–10 min) for antioxidants can increase bond strength in restored teeth after bleaching, indicating that the short application of antioxidants can reduce free radicals that affect bleaching on dentin bonding [37,51,52]. However, the current study clearly showed a relationship between the duration of application of antioxidant materials and the reduction in dissolved oxygen; specifically, we showed that applying antioxidant materials for 48 h can minimize or even almost eliminate peroxide residues on teeth after bleaching.

The mean differences in and distributions of dissolved oxygen between intervention groups at 1 h, 24 h, and 48 h were tested with a one-way ANOVA test ($p < 0.05$) that

showed that there were significant differences in the mean dissolved oxygen among sodium ascorbate 10% combined with Tween 80 0.2%, *Trigona* spp. propolis 10%, and the control (without antioxidants). However, running Tukey's HSD test ($p < 0.05$: significant difference) between the groups, i.e., *Trigona* spp. propolis 10%, sodium ascorbate 10% combined with Tween 80 0.2%, and the control, at observation times of 1 h, 24 h, and 48 h showed no significant difference between dissolved oxygen in the *Trigona* spp. propolis 10% and 10% sodium ascorbate combined with Tween 80 0.2% groups. This result is not in line with the opinion of Abhishek Parolia et al. (2010) that the antioxidant properties of propolis allow it to absorb free radicals and protect against gamma radiation better than vitamin C (ascorbic acid) [53]. However, studies conducted by S.A. El Sohaimy et al. (2014) compared propolis from China and Egypt with ascorbic acid and showed that ascorbic acid was more effective than propolis at low concentrations whereas, at high concentrations, propolis showed better antioxidant activity than ascorbic acid [54]. Propolis, rich in flavonoids, phenolics, and tannins, has good antioxidant properties through its ability to donate hydrogen from its hydroxyl group and its ability to donate electrons to resist the production of free radicals as a result of oxidative stress [55,56]. Meanwhile, the antioxidant effect of ascorbate is derived from its ability to react directly with a broad spectrum of reactive oxygen species, stop the chain reactions initiated by free radicals through electron transfer, and participate in the regeneration of the antioxidant properties of fat-soluble vitamin E [56]. Propolis has 1,1-diphenyl-2-picrylhydrazyl (DPPH) radical-scavenging activity that is higher than that of ascorbic acid due to propolis containing ascorbic acid in addition to flavonoids and phenolics that act as antioxidants [25,56]. A study reported the absence of a significant difference in the microtensile bond strength in teeth that had been bleached with 35% carbamide peroxide and were exposed to propolis or 10% sodium ascorbate as antioxidants [52]. However, the results of this study showed that the mean microtensile bond strength of propolis was higher than that of ascorbic acid [52].

Based on the current study, the results suggested that using propolis for 48 h successfully diminished peroxide remnants after an intra-coronal bleaching treatment which, in turn, made it possible to improve composite bonding and reduce waiting periods for composite restorations after the bleaching treatment. A limitation of this study is that a restoration strength test was not carried out after the applications of bleaching agents and antioxidants. Finally, further studies are needed for clinical applications, with larger sample sizes and varied follow-up periods to strengthen the evidence for the depletion of residual peroxides produced after intra-coronal bleaching through the application of antioxidant agents obtained from *Trigona* spp. propolis.

## 5. Conclusions

After conducting this study, the statistical results suggested that there was no significant difference between the effectiveness of sodium ascorbate 10% combined with Tween 80 0.2% and *Trigona* spp. propolis 10% to reduce peroxide residues after an intra-coronal bleaching treatment using hydrogen peroxide 35%. The lowest peroxide residue amount was found with the application of antioxidants for 48 h after the bleaching treatment. As a result, we believe that *Trigona* spp. propolis 10% has the ability to lessen the amount of peroxide left after intra-coronal bleaching treatments using 35 percent hydrogen peroxide, which could improve composite bonding and shorten the time needed to prepare for composite restorations after bleaching treatments.

**Author Contributions:** Conceptualization, A.C.T.; methodology, A.C.T.; validation, M.R.; data curation, S.S.; investigation, A.C.T.; writing—original draft preparation, A.C.T.; writing—review and editing, A.C.T. and S.S.; supervision, Y.-K.S. All authors have read and agreed to the published version of the manuscript.

**Funding:** This research received no external funding.

**Institutional Review Board Statement:** This study protocol was approved by the Ethics Committee Dentistry Faculty of Hasanuddin University with protocol code UH17120189 in 2019.

**Informed Consent Statement:** Not applicable.

**Data Availability Statement:** Data are present in the article.

**Acknowledgments:** The authors would like to thank Nurlindah Hamrun and Keng-Liang Ou for technical support.

**Conflicts of Interest:** The authors declare no conflict of interest.

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
