# Peer review of "The Potential of Trigona spp. Propolis as an Antioxidant Agent to Reduce Residual Peroxide after Intra-Coronal Bleaching Treatments"

_applsci, doi:10.3390/app12146996_

Round 1
Reviewer 1 Report
Please correct the details in the discussion section using new references
Reviewer 2 Report
Dear Author, I reviewed the manuscript (applsci-1789935) entitled The Potential of Trigona spp Propolis as an Antioxidant Agent to Reduce Residual Peroxide after Intra-coronal Bleaching Treatment. This manuscript presents relevant information about the antioxidant potential of propolis. However, some sections of the presented data can be improved. For this reason, I considered that this manuscript needs minor changes for being considered for its publication in this journal.
Additional comments.
Highlight the advantages of using propolis for bleaching dental treatment.
Check paragraph extension in this manuscript.
Include an experimental design that contains statistical factors and variables of response in the statistical analyses applied to the findings of this research.
Include a possible antioxidant mode of action of propolis bioactive compounds against free radicals.
Try to compare the obtained findings with similar assays where propolis or similar compounds were used to bleach dental materials.
Include future trends to keep working with the obtained data.
Try to indicate replicates per assay and standard deviation or statistical error in the results.
Try to conclude with a general statement of the most relevant part of this study.
Reviewer 3 Report
The manuscript presents the results obtained by using the propolis produced by Trigona spp. as an antioxidant to reduce residual peroxide after intra-coronal bleaching treatment. Authors used in their studies biological materials resulting from current stomatological surgical procedures. In my opinion, this manuscript contain too small amounts of original scientific data; the authors made different treatment on inert biologic materials, and made only one type of measurement (i.e. quantity of dissolved oxygen at three intervals of time). The resulted data were interpreted statistically and...that is all. Missing the ethical considerations, missing experimental data, missing new bibliographic references (in the manuscript I found only a reference from 2016, the references from the period 2017-2022 are missing. For these reasons, I consider that this article is not suitable for publication in this form.
Round 2
Reviewer 3 Report
In my opinion in this article experimental data are insufficient. More extensive studies must be made; for example, studies performed in vitro on normal cell lines, regarding the influence of proposed treatment on these.
